# *Rosa roxburghii* Fruit Extracts Upregulate Telomerase Activity and Ameliorate Cell Replicative Senescence

**DOI:** 10.3390/foods13111673

**Published:** 2024-05-27

**Authors:** Yan Huang, Haoyue Peng, Yifan Wu, Shengcheng Deng, Fahuan Ge, Wenbin Ma, Xue Zhou, Zhou Songyang

**Affiliations:** 1MOE Key Laboratory of Gene Function and Regulation, Guangzhou Key Laboratory of Healthy Aging Research, School of Life Sciences, Sun Yat-sen University, Guangzhou 510275, China; penghy8@mail2.sysu.edu.cn (H.P.); wuyf97@mail2.sysu.edu.cn (Y.W.); dengshch6@mail.sysu.edu.cn (S.D.); mawenbin@mail.sysu.edu.cn (W.M.); songyanz@mail.sysu.edu.cn (Z.S.); 2School of Pharmaceutical Sciences, Sun Yat-sen University, Guangzhou 510006, China; gefahuan@mail.sysu.edu.cn

**Keywords:** telomerase, *Rosa roxburghii* fruit, cell senescence

## Abstract

Anti-aging functional foods benefit the elderly. Telomeres are chromosomal ends that maintain genome stability extended by telomerase catalytic subunit TERT. Due to the end-replication problem, telomeres shorten after each cell cycle without telomerase in most human cells, and eventually the cell enters the senescence stage. Natural products can attenuate the aging process by increasing telomerase activity, such as TA-65. However, TA-65 is expensive. Other Chinese natural products may achieve comparable effects. Here, we found that *Rosa roxburghii* fruit extracts effectively increase TERT expression and telomerase activity in cultured human mesenchymal stem cells. Both *R. roxburghii* fruit extracts obtained by freeze-drying and spray-drying increased the activity of telomerase. *R. roxburghii* fruit extracts were able to reduce reactive oxygen species levels, enhance superoxide dismutase activity, and reduce DNA damage caused by oxidative stress or radiation. *R. roxburghii* fruit extracts promoted cell proliferation, improved senescent cell morphology, delayed replicative cellular senescence, attenuated cell cycle suppressors, and alleviated the senescence-associated secretory phenotype. Transcriptome and metabolic profiling revealed that *R. roxburghii* fruit extracts promote DNA replication and telomere maintenance pathways and decrease triglyceride levels. Overall, we provide a theoretical basis for the application of *R. roxburghii* fruit as an anti-aging product.

## 1. Introduction

Aging is an inevitable issue in the process of life development. With increasing age, the risk of disease and mortality continually increases [1]. The worsening of the aging population in contemporary society has led to a rising burden on medical expenditures. Effectively preventing and treating age-related chronic diseases and improving the health and quality of life of the aging population are important for human health and societal development.

Aging is a complex process that is related to molecular, cellular, and systemic alterations. In 2023, twelve aging hallmarks were proposed, including genomic instability, telomere attrition, epigenetic alterations, loss of proteostasis, impaired autophagy, nutrient sensing dysregulation, mitochondrial dysfunction, cellular senescence, stem cell exhaustion, altered intercellular communication, chronic inflammation, and metabolic dysfunction [2]. Stem cell exhaustion is one of the twelve indicators of aging, highlighting the importance of research related to aging in stem cells. Stem cells play a vital role in maintaining tissue homeostasis and promoting tissue repair.

Mesenchymal stem cells (MSCs) are multipotent cells found in various tissues, such as bone, umbilical cord blood, and dental pulp. They can differentiate into osteoblasts, adipocytes, fibroblasts, etc. MSCs can spontaneously migrate to damaged areas, differentiate into the required tissue, secrete anti-inflammatory cytokines, and modulate immunology to perform regenerative functions and exert therapeutic effects [3]. To date, MSCs are considered to be one of the most promising types of stem cells for cell therapy [4]. The proliferative and differentiation potential of MSCs decreases with increasing in vitro culture passages, limiting their therapeutic effects [5]. Morphological and functional changes occur in aging MSCs, including cell enlargement, loss of spindle-shaped characteristics, skewed differentiation, and decreased cell clonogenicity [6]. Aged MSCs also influence adjacent cells through the secretion of senescence-associated secretory phenotype (SASP) factors, promoting inflammation and aging [6]. Extracellular vesicles derived from MSCs exhibit anti-inflammatory, tissue-repair-promoting, and antifibrotic effects [7]. The production of extracellular vesicles from aging MSCs increases, and the composition of vesicle contents changes, possibly further accelerating the aging of adjacent cells [8].

Telomeres are long repetitive sequences located at the ends of eukaryotic chromosomes that play a crucial role in maintaining genome stability and functional integrity. Due to the end-replication problem, a small segment of the sequence cannot be replicated with each replication of the DNA end. Therefore, with each round of replication, cells lose a portion of telomeric DNA, and ultimately, when telomeres shorten to a threshold length, cells undergo senescence known as the “Hayflick limit” [9,10]. Telomere shortening accelerates the aging process, and there is a negative correlation between telomere length and mortality in people older than 60 years [11]. Research has shown that telomere length is positively correlated with a healthy lifestyle and negatively correlated with the risk of various age-related diseases, such as cardiovascular diseases, obesity, and chronic pain [12,13]. Telomere maintenance, which is achieved through two main mechanisms, is essential for the progression of cancer. Approximately 85–90% of human tumor cells maintain telomere length through telomerase, while the remaining 10–15% use the alternative lengthening of telomere (ALT) pathway [10].

The replication capacity of human somatic cells is limited. A critically short telomere triggers the DNA damage response (DDR) pathway, which is sensed by ATM and ATR kinases, as indicated by DNA damage markers such as γH2AX and mediated by the transcription factor p53. p53 positively regulates p21 expression, which inhibits CDK2/cyclin E kinase activity, leading to the dephosphorylation of pRb, causing cell cycle arrest. In addition to the p53/p21/pRb signaling pathway, p16-Rb also plays a significant role in regulating cell senescence by inhibiting the activity of kinases CDK4 and CDK6, promoting the inhibitory effect of the E2F-Rb transcription factor on downstream genes [14].

Telomeres are rich in guanine and susceptible to oxidative stress damage [15]. Under oxidative stress conditions, single-strand breaks accumulating on telomeres can cause replication fork stalling and the incomplete replication of chromosome ends, leading to telomere shortening [16]. The cellular aging process mediated by telomere DNA shortening can be inhibited by enhancing the ability of the cell to resist oxidative damage [17]. In addition, telomere shortening increases the sensitivity of chromosomes to radiation, leading to an increased probability of chromosomal aberrations caused by radiation [18].

Natural products can influence aging-related metabolic pathways, provide health benefits, and extend lifespan. *Astragalus membranaceus* alleviates oxidative stress reactions in cerebral ischemia rats by reducing reactive oxygen species (ROS) and malondialdehyde (MDA) levels [19]. *A. membranaceus* water extract can suppress cytokine production by inhibiting the NF-κB signaling pathway, alleviating inflammatory responses [20]. Ethanol extracts of *Astragalus mongholicus* roots induce significant telomere elongation in lymphocytes [21]. TA-65, a compound containing cycloastragenol isolated from *A. membranaceus* roots, has been shown to increase telomerase activity in embryonic fibroblasts, inhibit DNA damage, and enhance immune function [22,23]. Ginseng is a valuable traditional Chinese medicine that contains ginsenosides, which are among the most important active ingredients for its anti-aging function [24].

*R. roxburghii* fruit is a valuable wild resource on the Yun–Gui Plateau and Western Sichuan Plateau and is a key economic plant in Guizhou Province. *R. roxburghii* fruit is rich in ascorbic acid (vitamin C), which has significant potential in epigenetics, antioxidation, anticancer, and stem cell reprogramming [25,26,27]. Among the different varieties of cultivated *R. roxburghii* fruit, Guinong No. 5 exhibits high superoxide dismutase (SOD) activity, a high level of vitamin C, and a high antioxidant capacity [28].

In terms of its anti-aging function, *R. roxburghii* fruit significantly has reduced total cholesterol and triglyceride levels [29]. *R. roxburghii* fruit polysaccharides have been found to regulate mouse intestinal flora, thereby regulating intestinal barrier damage and inflammation caused by a high-fat diet [30]. Experiments in D-galactose-induced aging mouse models showed that *R. roxburghii* fruit can delay the aging of organs and skin [31]. In fruit flies, *R. roxburghii* fruit can significantly extend the average lifespan of both male and female flies, suggesting its enormous potential for anti-aging [32]. The flavonoid components of *R. roxburghii* fruit can also serve as radioprotective agents [33].

In order to address the role and mechanism of *R. roxburghii* fruit, this study explored, for the first time, the relationship between *R. roxburghii* fruit extracts and telomerase activity. The role of *R. roxburghii* fruit extracts in telomerase regulation complements the mechanism of *R. roxburghii* fruit’s anti-aging function. Using telomerase activity as an indicator, we identified both freeze-drying and spray-drying extraction methods that maintain efficacy, providing an important reference for *R. roxburghii* fruit extract production processes. Therefore, we can further research and develop effective anti-aging agents by combining *R. roxburghii* fruit with other natural products, maximizing the utilization and effectiveness of *R. roxburghii* fruit and promoting the development of related industries.

## 2. Materials and Methods

### 2.1. Extraction Process

*R. roxburghii* fruit extracts of Gui Nong No.5 (CL-1) or wildtype (CL-2) were extracted as follows. The fruits were washed, removed with seeds, cut into pieces, mixed with water (1:5 to 1:30), and homogenized using a rapid homogenizer (FLUKO Homogenization) for 30 min. Then, the mixture were treated with high-pressure disruption extraction (Guangzhou Juneng Biology Technology Co., Ltd., Guangzhou, China) with 100 mPa pressure at 4 °C. The samples were filtered, and liquids were treated through vacuum freeze-drying. For CL-3 and CL-4, the fruits were washed, the seeds were removed, and they were cut into pieces and then squeezed. The juice was vacuum freeze-dried (CL-3) or spray-dried (CL-4). CL-5 was made by CL-1 with polysaccharide depleted by ethanol precipitation.

### 2.2. Cell Lines

Human mesenchymal stem cells (hMSC) from the umbilical cord, HeLa human cervical cancer cells, and SK-MEL-2 human skin melanoma cells were obtained from commercial cell lines. Cells were cultured in DMEM (Corning, 10-013-CVR, New York, NY, USA) with 10% fetal bovine serum (HyClone, SH30084.03, North Logan, UT, USA) in 5% CO_2_ at 37 °C.

### 2.3. Quantitative Telomeric Repeat Amplification Protocol (Q-TRAP)

Q-TRAP assays were performed as described. Briefly, 10^5^ cells were lysed on ice for 30 min in 100 μL of NP40 lysis buffer (10 mM Tris-HCl (pH 8.0), 1 mM MgCl_2_, 1 mM EDTA, 0.25 mM sodium deoxycholate, 150 mM NaCl, 1% NP-40, 10% glycerol, and 1% fresh protease inhibitor cocktail) and then centrifuged at 13,200 rpm for 10 min at 4 °C. The protein concentration of the supernatant was measured by a BCA kit (Thermo Scientific, 23225, Waltham, MA, USA). A 1 μg sample was mixed with 100 ng/μL TS primer, 100 ng/μL ACX primer, and 1 mM EGTA in 2 × RealStar Green Power Mixture (with ROX) (GeneStar, Beijing, China) and then incubated at 30 °C for 30 min for telomeric repeat extension and PCR amplification (40 cycles, 95 °C for 15 s, and 60 °C for 60 s) using the Step One PlusTM Real-Time PCR system (Applied Biosystems, Waltham, MA, USA). Series-diluted 293T samples were used for standard curve.

### 2.4. SA-β-Gal Staining

The assay was performed by using a Senescence β-Galactosidase Staining Kit (Beyotime, C0602, Shanghai, China). Briefly, cells were seeded in 12-well plates and fixed with 4% formaldehyde for 15 min at room temperature. The fixed cells were then washed with PBS 3 times and incubated with fresh SA-β-gal staining reagent mixture containing 1.0 mg/mL X-galactosidase at 37 °C for 24 h for the microscopic observation.

### 2.5. Immunofluorescence

The cells were seeded on to cover slips in 12-well plates and incubated overnight with 500 μL of gelatin per well. The gelatin in the plate was removed, the density of each well was 50%, and cells was treated with or without 100 μg/mL *R. roxburghii* fruit extracts, or 100 μM H_2_O_2_, or 2 Gy X-Ray according to the experimental requirements. The supernatant was discarded, and the cells were washed twice with PBS and placed on ice for 5 min. Then, 500 μL of fixing solution was added to each well, and the plates were placed on ice for 30 min. The fixing solution was discarded, and the plates were washed with PBS. One milliliter of sealing solution was added to each well, and the plate was sealed at room temperature for 1 h (the sealing solution was PBST containing 5% goat serum, and the Tween 20 ratio was 0.5%). The γ-H2AX antibody (CST, 9718S, Boston, MA, USA) was diluted with a blocking solution and added to the cover slips overnight. The primary antibody was discarded, and cover slips were gently washed with PBST 3 times. The cover slips were then incubated with secondary antibody anti-Rabbit 488 nm (Invitrogen, A-21206, Waltham, MA, USA) for 1 h at room temperature and washed with PBST 3 times. The cover slip was stained with Hochest33342, added with mounting medium, and then the edge of the cover slip was fixed with a small amount of nail polish. All samples were visualized on a LEICA microscope equipped with appropriate filters.

### 2.6. ROS and SOD Activities

ROS was measured by the ROS Assay Kit (Beyotime, S0033S, Shanghai, China). Cells were loaded with DCFH-DA diluted 1:1000 in serum-free medium (BASO, A2501P05C, Zhuhai, China). After 37 °C incubation for 20 min, cells were washed 3 times with serum-free medium. The cells were trypsinized and analyzed using FACs with 488 nm as the excitation wavelength and 525 nm as the emission wavelength.

Total SOD activity was detected by a Total Superoxide Dismutase Assay Kit with NBT (Beyotime, S0109, Shanghai, China). Superoxide anion was produced by the reaction system of Xanthine and Xanthine Oxidase (XO). Nitrogen blue tetrazole (NBT) was reduced to blue formazan, which has strong absorption at 560 nm. SOD can clear superoxide anion, thus inhibiting the formation of formazan. The darker the blue of the reaction solution, the lower the activity of superoxide dismutase, and vice versa. The activity level of superoxide dismutase can be calculated by colorimetric analysis. Cells were scraped at 4 °C in PBS and washed 3 times. Cells were lysed by an electric homogenizer, the supernatants were collected, and the protein concentration was measured using a BCA kit (Thermo Scientific, 23225, Waltham, MA, USA). Samples were diluted in 20 μg/mL. A total of 20 μL of sample, 160 NBT/enzyme working solution, and 20 μL of reaction starting solution were incubated at 37 °C for 30 min. Absorbance at 560 nm was detected. The activity of total SOD was calculated according to the formula in the manufacturer’s instructions.

### 2.7. DNA Damage Induced by H_2_O_2_ or Irradiation

To induce a cell morphology change, MSCs were pretreated with *R. roxburghii* fruit extract or vitamin C for 48 h, treated with 200 μΜ H_2_O_2_ for 4 h, and then the relative ROS level was analyzed. To induce DNA damage by H_2_O_2_, MSCs were pretreated with *R. roxburghii* fruit extract for 24 h and then treated with 100 μΜ H_2_O_2_ for 2 h. γH2AX immunofluorescence was performed. To induce DNA damage by irradiation, MSCs were pretreated with 100 μg/mL *R. roxburghii* fruit extract for 24 h and then treated with 2 Gy of X-rays (Xstrahl Life Sciences, CIX3, Suwanee, GA, USA). After 30 min of incubation in 5% CO_2_ at 37℃, the cells were fixed and immunostained with a γH2AX antibody.

### 2.8. Western Blot

The cells were lysed in RIPA buffer (Beyotime, P0013B), and the protein concentration was determined by a BCA kit (Thermo Scientific, 23225). Proteins of 80 μg were separated by SDS-PAGE, transferred to PVDF membranes, and probed with the indicated primary antibody and the corresponding second antibody by the Odyssey Infrared Imager (LI-COR). Information about the antibodies is as follows. Anti-p16 (BD, 554079), anti-p21 (Merck Millipore, OP64, St. Louis, MI, USA), anti-p53 (Santa Cruz Biotechnology, sc-126, Dallas, TX, USA), anti-TERT (Abcam, Ab32020, Cambridge, UK), anti-Mouse 680 nm (LI-COR, 926-68070, Lincoln, OR, USA), and anti-Rabbit 800 nm (Odyssey, IRDye-800, Lincoln, OR, USA).

### 2.9. Cell Growth Curve Analysis

The counting plate was wiped with alcohol, and the cover glass was covered. After the cells were digested and diluted, 10 μL of cells was slowly added to the upper and lower grooves, respectively. The number of cells in four diagonal squares was observed and recorded under a microscope, the number of cells in a total of eight squares in the upper and lower grooves was recorded as N, and the cell concentration was (N/8) × 10^4^ cells/mL. The cells in each group were repeatedly counted three times, the average number of cells was calculated as Nf, and the initial number of seeded cells was designated as Ni. After each count, 10^5^ cells were inoculated into a new six-well plate, and after the cells were attached to the wall, the medium was changed, and *R. roxburghii* fruit extract or vitamin C was added. The growth curve was fitted according to the formula: PD = log_2_(Nf/Ni).

### 2.10. RNA-Seq and Metabolic Profiling

RNA-seq was performed by Geneplus Technology using the DNBSEQ-T7 platform. Total RNA was extracted, and the PCR library was constructed by the standard method. The prepared DNA Nanoball (DNB) was loaded onto the Patterned Array and sequenced using Combinatorial Probe Anchor Synthesis (cPAS). After the sequencing primer anchor molecules and fluorescent probes were polymerized on DNA nanospheres, a high-resolution imaging system was used to collect, read, and recognize the light signal to obtain the single base sequence information. In the next cycle, more base sequence information was obtained, and finally the original sequencing data were obtained through multiple cycles. A double-ended PE150 sequencing strategy was used. The RNA-seq data were uploaded to the NCBI Gene Expression Omnibus database (GSE261987).

For metabolic profiling, cells were scraped at 4 °C in PBS and washed 3 times. Cell pellets were stored at −80 °C.

For hydrophilic compounds, the sample stored at −80 °C was thawed on ice. A 500 μL solution (methanol–water = 4:1, *v*/*v*) containing an internal standard was added to the cell sample and vortexed for 3 min. The sample was placed in liquid nitrogen for 5 min and on dry ice for 5 min and then thawed on ice and vortexed for 2 min. This freeze-thaw cycle was repeated three times. The sample was centrifuged at 12,000 rpm for 10 min (4 °C). A 300 μL aliquot of the supernatant was collected and placed at −20 °C for 30 min. The sample was then centrifuged at 12,000 rpm for 3 min (4 °C). Two hundred microliter aliquots of the supernatant were transferred for LC-MS analysis. The sample extracts were analyzed using an LC-ESI-MS/MS system (UPLC, ExionLC AD, https://sciex.com.cn/; MS, QTRAP^®^6500+ System, https://sciex.com/, accessed on 15 January 2024). LIT and triple quadrupole (QQQ) scans were acquired on a triple quadrupole-linear ion trap mass spectrometer (QTRAP), QTRAP^®^ LC-MS/MS system, equipped with an ESI Turbo Ion–Spray interface, operating in positive- and negative-ion mode and controlled by Analyst 1.6.3 software (Sciex).

For the analysis of the hydrophobic compounds, the samples were removed from the −80 °C freezer and thawed on ice. One milliliter of extraction solvent (Methyl tert -butyl ether:MeOH = 3:1, *v*/*v*) containing an internal standard mixture was added. After the mixture was allowed to warm for 15 min, 200 μL of ultrapure water was added. The mixture was vortexed for 1 min and centrifuged at 12,000 rpm for 10 min. Then, 500 μL of the upper organic layer was collected and evaporated using a vacuum concentrator. The dry extract was dissolved in 200 μL of reconstituted solution (acetonitrile–iso-propyl alcohol = 1:1, *v*/*v*) for LC-MS/MS analysis. The sample extracts were analyzed using an LC-ESI-MS/MS system and analyzed with the same instrument and methods as the hydrophilic compounds.

### 2.11. Statistical Analysis and Quantification

The data are shown as the means ± SDs. The experiments were carried out with three technical replicates. Student’s *t* test and a one-way ANOVA were used for statistical significance analyses with the software GraphPad Prism version 6.0 (San Diego, CA, USA). The fitting curves were depicted using Original version 9.0. A p value less than 0.05 indicated statistical significance (* *p* < 0.05, ** *p* < 0.01, *** *p* < 0.001, **** *p* < 0.0001). The quantification of the IF results and β-gal staining results was statistically analyzed using Image J bundled with 64-bit Java 8. FACs data were analyzed with the Software CytExpert for editing and exporting.

## 3. Results

### 3.1. R. roxburghii Fruit Extract Can Increase hMSC Telomerase Activity

To broaden the scope of the anti-aging mechanism and further investigate the impact of *R. roxburghii* fruit extraction on its anti-aging effect, we obtained five *R. roxburghii* fruit extracts with different processing methods and tested their effects on telomerase activity in hMSCs (Figure 1A). All the samples were tested at two concentrations, with a low concentration of 50 μg/mL (L) and a high concentration of 100 μg/mL (H). *R. roxburghii* fruit extracts contain approximately 10% vitamin C (Vc), which could serve as a positive control in the experiment. We established low- and high-concentration groups with final concentrations of 5 μg/mL and 10 μg/mL, respectively.

Among the five *R. roxburghii* fruit extracts, the sample with the code CL-3 showed the greatest increase in telomerase activity. Compared with the low-concentration group, the high-concentration group exhibited a superior effect, and the results from the three independent replicates consistently showed the ability of the high-concentration group to stably increase telomerase activity in hMSCs. In addition to CL-3, CL-4 also increased telomerase activity. Furthermore, 10 μg/mL Vc promoted an increase in telomerase activity in hMSCs (Figure 1B). Consistent with telomerase activity, the TERT mRNA level was also elevated upon CL-3 treatment (Figure 1C). Both short-term and long-term CL-3 treatment upregulated the TERT protein levels (Figure 1D).

For safety considerations, we tested the impact of CL-3 on telomerase activity in tumor cells. At concentrations of 25/50/100/200 μg/mL, telomerase activity did not increase in human melanoma cells or human cervical cancer cells (Appendix A). This, to some extent, proves that CL-3, when telomerase activity increases in human mesenchymal stem cells, does not simultaneously increase telomerase activity in tumor cells.

Since the extraction process used for CL-3 may differ from that used for the commercial extraction of *R. roxburghii* fruit, CL-3 could be effective at increasing telomerase activity. To further investigate whether the advantages of the CL-3 extraction process are generally applicable, we selected *R. roxburghii* fruit extracts from three different commercial companies. The properties of the samples from the three companies are shown in Appendix A. Among them, the appearance, odor, and solubility of the sample from Company c were most similar to those of CL-3 (Appendix A). The effects of the three commercial samples on regulating telomerase activity in hMSCs were tested, and the results showed that the fold increase in the telomerase activity of the Company C sample was basically consistent with that of CL-3 (Appendix A). The sample from Company c was produced by spray-drying. This result is consistent with that of CL-4, which was also produced by spray-drying and upregulated telomerase activity (Figure 1A,B). This demonstrates that both the freeze-drying and spray-drying processes can preserve the active ingredients of *R. roxburghii* fruit that can modulate telomerase activity.

To identify the key active ingredients in CL-3 that contribute to the increase in hMSC telomerase activity, we conducted tests on three *R. roxburghii* fruit components obtained commercially: oleanolic acid, *R. roxburghii* fruit glycoside, and quercetin. Concentrations were set based on the known literature. None of them activated telomerase activity in hMSCs (Appendix A), preliminarily excluding the functional impact of these *R. roxburghii* fruit components on telomerase.

### 3.2. R. roxburghii Fruit Extract Enhances the Antioxidative Capacity of hMSCs

MSC aging is driven by various events. Along with cell proliferation, irreversible cell cycle arrest, changes in cell morphology, and impaired differentiation ability can occur due to telomere shortening, the accumulation of reactive oxygen species (ROS), and DNA damage [34]. ROS can induce protein denaturation and cross-linking, leading to the loss of the biological activity of many enzymes and hormones within the body. It can also disrupt nucleic acid structures, resulting in abnormal metabolism [35]. The level of ROS can be assessed to evaluate cellular oxidative stress damage. After treating cells with CL-3 for 48 h and loading the DCFH-DA probe, flow cytometry was used to detect fluorescence signals. The fluorescence signal peak of the 100 μg/mL CL-3 treatment group shifted to the left on the graph and was obviously different from that of the control group. This indicates an overall reduction in the fluorescence signal intensity in the CL-3 treatment group (Figure 2A). The statistical analysis of FITC values for 10,000 cells in each group showed that both high and low concentrations of CL-3 significantly reduced the fluorescence signal intensity compared to that in the control group, indicating that CL-3 can lower the overall ROS levels in cells (Figure 2B, **** *p* < 0.0001).

SOD is an antioxidative metal enzyme present in organisms. It catalyzes the dismutation of superoxide anions into oxygen and hydrogen peroxide, playing a crucial role in the balance of oxidative and antioxidative processes in the body [36]. The effect of CL-3 treatment on cell SOD activity was detected based on the NBT colorimetric reaction. Vitamin C could partially increase hMSC SOD activity, although the statistical analysis showed no significant difference. Notably, both 50 μg/mL and 100 μg/mL CL-3 significantly increased cell SOD activity (Figure 2C, * *p* < 0.05, ** *p* < 0.01).

Hydrogen peroxide (H_2_O_2_) stimulation causes acute oxidative damage to cells, simulating oxidative stress-induced imbalance under natural conditions and further inducing cell damage and apoptosis. As expected, stimulation with 200 μM H_2_O_2_ for 4 h changed the cell morphology, with the cells becoming round and detaching from the dish, indicating severe damage and cell death. However, pretreatment with CL-3 for 48 h maintained the original state of the cells without damaging the cells caused by H_2_O_2_ stimulation (Figure 2D). The detection of total ROS levels in cells after H_2_O_2_ stimulation using a fluorescent probe revealed that H_2_O_2_ caused acute oxidative damage, increasing intracellular ROS levels. Pretreatment with vitamin C or CL-3 enhanced the antioxidative capacity of the cells, reducing the increase in total intracellular ROS caused by acute damage. Moreover, CL-3 had a greater enhancing effect than vitamin C (Figure 2E).

### 3.3. R. roxburghii Fruit Extract Alleviates the DNA Damage Response in hMSCs

Oxidative stress can increase the levels of intracellular ROS, which are among the most common inducers of DNA damage in cells, leading to cell cycle arrest and cell senescence. After 100 μM H_2_O_2_ oxidation, the fluorescence signal of γH2AX in the control group cells increased, indicating a robust DNA damage signal. In the CL-3 pretreatment group, the number of γH2AX foci was significantly lower than that in the control group with H_2_O_2_ stimulation, indicating that CL-3 pretreatment could reduce the extent of DNA oxidation damage (Figure 3A, **** *p* < 0.0001). Similar results were obtained in X-ray-irradiated cells. X-ray ionizing radiation, which can directly break chemical bonds between nucleotides and induce DNA double-strand breaks and nucleotide base cross-linking, is detrimental to cells. γH2AX staining showed that X-ray irradiation induces obvious DNA damage signals. Pretreatment with *R. roxburghii* fruit extract significantly alleviated γH2AX signals induced by X-ray irradiation in hMSCs (Figure 3B, **** *p* < 0.0001).

### 3.4. R. roxburghii Fruit Extract Alleviates Replicative Senescence in hMSCs

Then, we cultured hMSCs from PD8 cells supplemented with CL-3 or vitamin C and passaged the cells every 3 days. After each count, 1 × 10^6^ cells were reseeded into new culture dishes. Cumulative population doubling was plotted, indicating that *R. roxburghii* fruit extract promotes cell proliferation (Figure 4A). Senescence-associated β-galactosidase (SA-β-gal) activity is an important indicator of cell senescence. Long-term treatment with 100 μg/mL CL-3 for 30 days (PD19) inhibited the accumulation of SA-β-gal signals, while there was no significant difference between the vitamin C treatment group and the control group (Figure 4B).

The senescence-associated secretory phenotype (SASP) is a crucial hallmark of senescent cells. The SASP affects communication between cells, mediates pathological development, and influences the aging process. Four representative SASP factors were selected, including the proinflammatory cytokine INF-γ, the chemokines IL-8 and CXCL-10, and the matrix metalloproteinase MMP2. After 30 days of in vitro culture, the mRNA levels of SASP factors in the CL-3 treatment group were significantly lower than those in the control group (Figure 4C, * *p* < 0.05, ** *p* < 0.01).

Replicative senescence can cause permanent cell cycle arrest due to telomere shortening and DNA damage, which are mainly mediated by the ATM/ATR kinases and p53/p21/pRb signaling pathways. The transcription factor p53 activates p21 at the transcriptional level. After activation, p21 inhibits CDK2/cyclin E kinase activity, leading to the dephosphorylation of pRb and cell cycle arrest. The p16-Rb pathway regulates the cell cycle by inhibiting CDK4 and CDK6 kinase activity. The expression levels of p16/p21/p53 increase with cell senescence. By examining the long-term effects of CL-3 and vitamin C on the expression levels of p16/p21/p53, it was found that with increasing culture time, the mRNA levels of p16/p21/p53 increased after 30 days. CL-3 and vitamin C reduced the increase in the expression of p16/p21 caused by cell senescence, and the effect of CL-3 was greater than that of vitamin C. However, CL-3 had no significant impact on p53 mRNA after 30 days (Figure 4D). At the protein level, long-term CL-3 and vitamin C treatment for 30 days reduced the protein expression levels of p16/p21/p53 (Figure 4E). In summary, CL-3 can delay the replicative senescence of hMSCs under in vitro culture conditions.

### 3.5. R. roxburghii Fruit Extract Targets Multiple Pathways and Metabolically Modulates Rejuvenation

To further explore the mechanisms underlying the ability of CL-3 to delay replicative senescence in hMSCs, we performed transcriptome sequencing in 24 h CL-3-treated cells. Three independent replicate samples for both the control group and the CL-3 treatment group were sequenced. Reactome-dot plots and Reactome-cent plots of the differentially expressed genes in *R. roxburghii* fruit extract-treated cells revealed that “TP53 regulates metabolic genes” and “peptide hormone metabolism” (Figure 5A,B). The KEGG-dot plot of differentially expressed genes showed “nitrogen metabolism” and “glycine, serine and threonine metabolism”, etc. (Figure 5C). The GSEA of the differentially expressed genes in *R. roxburghii* fruit extract-treated cells revealed that they were involved in DNA replication, telomere maintenance, and lymphocyte chemotaxis (Figure 5D). To validate the reliability of the sequencing data, four DEGs were selected for expression level verification. The RNA-seq data revealed that the significantly upregulated genes BIRC3, HMGCS1, and CCL2 were significantly upregulated at the mRNA level, and the downregulated gene SESN2 was also consistent with the RNA-seq analysis (Figure 5E, *** *p* < 0.001).

RNA-seq revealed metabolic changes in response to *R. roxburghii* fruit extract treatment; thus, we performed the metabolic profiling of both hydrophilic and hydrophobic compounds. KEGG enrichment of the differentially abundant metabolites in *R. roxburghii* fruit extract-treated cells revealed “lipid and atherosclerosis”, “cholesterol metabolism”, and “vitamin digestion and absorption” (Figure 5F). A total of 58 upregulated metabolites, such as traumatic acid, lysophosphatidylcholine (LPC), phe-hyp, and cGMP, were detected in CL-3-treated cells. The 280 downregulated metabolites included triglyceride (TG), ceramides (CER), and phosphatidylglycerol (PG) (Figure 5G). This result is consistent with a previous report showing that *R. roxburghii* fruit can significantly reduce total cholesterol and triglyceride levels [29]. We also analyzed the metabolic profile of cells treated with vitamin C. Compared with *R. roxburghii* fruit extract, vitamin C targets distinct metabolic pathways (Appendix A). Overall, *R. roxburghii* fruit extract metabolically modulates cell rejuvenation.

## 4. Discussion

Aging and health are complex and multifaceted topics influenced by genetic, environmental, psychological, and physiological factors [2]. With the advancement of medicine, the average human lifespan has been steadily increasing. However, as the population ages, the incidence of neurodegenerative diseases, cardiovascular diseases, diabetes, and other chronic conditions continues to increase, imposing significant burdens on society, the economy, and healthcare systems. To address this issue, there is growing interest in intervening in aging-related physiological processes and developing potential anti-aging products.

TA-65, a cycloastragenol-containing commercial product isolated from *A. membranaceus* root, can increase telomerase activity. It can elongate the lifespan of mice and promote human T cell function [22,23]. TA-65 is expensive. Natural substances known to exert anti-aging effects by enhancing telomerase activity may have limitations, such as high price, complex extraction processes, and patent-related issues. To achieve long-term development, exploring the abundant plant resources in China and researching localized anti-aging products align with national development needs. This approach could alleviate healthcare pressures, stimulate local economic development, and carry crucial significance.

Current research on *R. roxburghii* fruit is primarily concentrated in China, especially in the Guizhou region. Despite numerous products on the market utilizing *R. roxburghii* fruit as a primary ingredient, several aspects require further investigation. First, variations in different varieties of *R. roxburghii* fruit may exist, and diverse methods of extraction and purification will lead to significant differences in quality, necessitating the establishment of standardized preparation and extraction processes for maximized effectiveness. Second, the mechanism underlying its effects remains inadequately understood. A better understanding of the molecular mechanisms of *R. roxburghii* fruit will facilitate its application.

In light of these considerations, our study delved deeper into the anti-aging properties of *R. roxburghii* fruit. Through in vitro telomerase activity assays, we explored the impact of *R. roxburghii* fruit extracts from different sources on cellular telomerase activity. On this basis, we showed that both freeze-drying and spray-drying methods can preserve the telomerase-activating effect of *R. roxburghii* fruit extracts. The experimental results demonstrated that *R. roxburghii* fruit extract has the ability to enhance telomerase activity, strengthen cellular antioxidant defenses, reduce cell DNA damage caused by oxidative stress or irradiation and, consequently, alleviate cellular replicative senescence phenotypes. Transcriptome sequencing revealed that *R. roxburghii* fruit extract potentially promotes cellular proliferation and DNA repair-related pathways. Metabolic profiling supported the metabolic modulation of cell rejuvenation. These results were supportive of the reported metabolic regulation role of *R. roxburghii* [29]. This study provides possible directions for the further exploration of the molecular mechanisms underlying the anti-aging effects of *R. roxburghii* fruit extracts, laying a theoretical foundation for subsequent product development and applications.

However, this study has its limitations. Firstly, the active components of the *R. roxburghii* fruit extracts that affect telomerases in vivo are still unknown. *R. roxburghii* fruit contains vitamin C, SOD, polysaccharide, triterpenoids, such as oleanolic acid, *R. roxburghii* fruit glycoside, and quercetin, flavonoids, phenolics, organic acids, etc. We have preliminarily excluded polysaccharide (Figure 1), oleanolic acid, *R. roxburghii* fruit glycoside, and quercetin (Appendix A). Apart from vitamin C, SOD, flavonoids, phenolics, organic acids, etc., may be candidate active ingredients that activate telomerase in the extracts. More experiments are needed to test their role in telomerase regulation and their synergetic effects with vitamin C. Secondly, the current experimental data are limited on the cellular level, and validation through animal experiments is lacking. Some experiments can be performed, like feeding model animals such as rats with specific active ingredients of the *R. roxburghii* fruit extracts and using the rat’s serum to culture MSC to evaluate the telomerase regulating activity and anti-aging effect, as well as the long-term feeding-specific active ingredients of the *R. roxburghii* fruit extracts to mice and observing the anti-aging effect in vivo. These results will not only address the in vivo anti-aging role of the extracts but also illustrate the key active components in the extracts. The verification of only a subset of significantly differentially expressed genes raises the need for a more comprehensive examination. Regarding applications, the synergistic effect of CL-3 with other natural molecules in enhancing telomerase activity will open new avenues for formulation development. Nevertheless, the journey from basic research to product development is extensive. Considering the constraints of practical production conditions, the proportions of formulations, solubility of components, storage conditions, etc., require in-depth investigation. In future research, a closer integration of experimentation and application is needed to further elucidate the mechanisms and application value of CL-3, expand the scope of research, and comprehensively evaluate its utility and safety. This study provides a theoretical basis for the application of *R. roxburghii* fruit and lays the groundwork for the development of natural anti-aging products.

## 5. Conclusions

For the first time, we found that *R. roxburghii* fruit extracts upregulate TERT expression and telomerase activity in cultured hMSCs. Both *R. roxburghii* fruit extracts obtained by freeze-drying and spray-drying increased the activity of telomerase. *R. roxburghii* fruit extracts can reduce ROS levels, enhance SOD activity, and ameliorate DNA damage caused by oxidative stress or radiation. *R. roxburghii* fruit extracts promoted cell proliferation, improved senescent cell morphology, delayed replicative cellular senescence, and attenuated cell cycle suppressors and the SASP factors. Transcriptome and metabolic profiling revealed that *R. roxburghii* fruit extracts promote DNA replication and telomere maintenance pathways and decrease triglyceride levels. In summary, we provide a theoretical basis for the application of *R. roxburghii* fruit as an anti-aging natural product.

## Figures and Tables

**Figure 1 foods-13-01673-f001:**
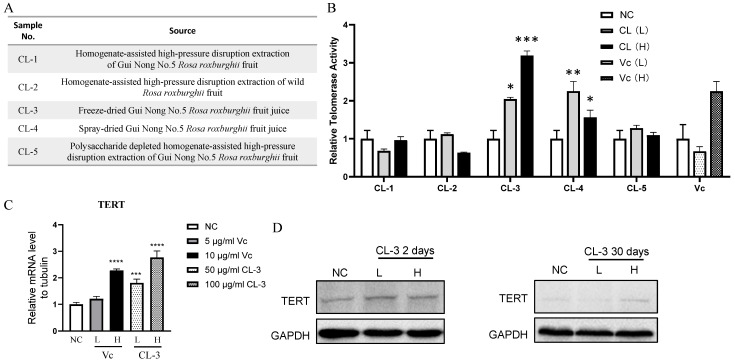
*R. roxburghii* fruit extract enhances telomerase activity and TERT expression in mesenchymal stem cells. (**A**,**B**) Human mesenchymal stem cells were treated with different *R. roxburghii* fruit extracts from (**A**) or vitamin C (Vc), and a quantitative PCR-based telomerase repeat-amplification protocol (Q-TRAP) was used to detect telomerase activity. PBS was used as a negative control (NC). For *R. roxburghii* extract, L, low concentration, 50 μg/mL; H, high concentration, 100 μg/mL. Student’s *t* test, * *p* < 0.05, ** *p* < 0.01, *** *p* < 0.001. (**C**) Relative mRNA level of TERT after treatment with *R. roxburghii* fruit extract or vitamin C. Student’s *t* test, *** *p* < 0.001, **** *p* < 0.0001. (**D**) Protein levels of TERT after treatment with *R. roxburghii* fruit extract or vitamin C for 2 or 30 days. The signals were detected by ECL. The error bars represent the means ± SDs, *** *p* < 0.001. GAPDH served as a loading control.

**Figure 2 foods-13-01673-f002:**
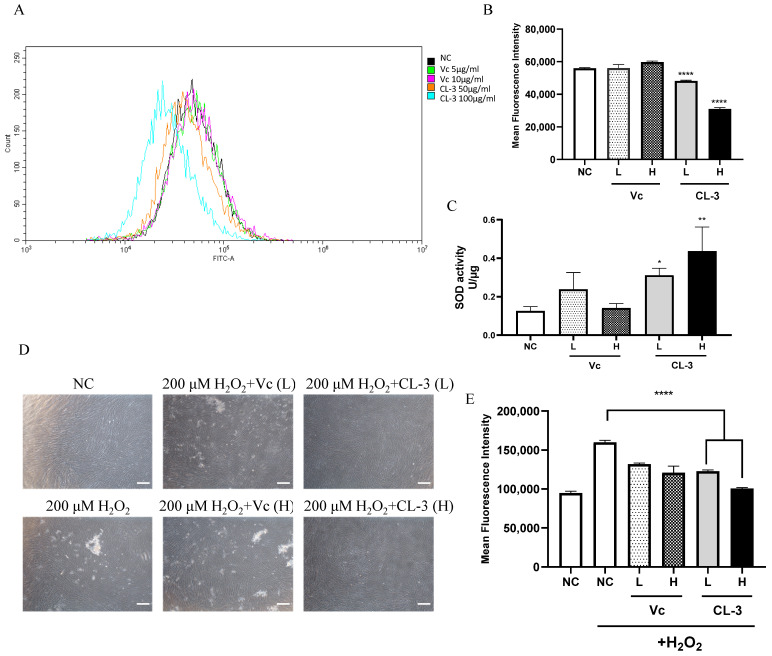
*R. roxburghii* fruit extract decreases reactive oxygen species levels, enhances cellular superoxide dismutase activity, and alleviates oxidative stress in mesenchymal stem cells. (**A**,**B**) Reactive oxygen species were measured with a Beyotime ROS Assay Kit using a DCFH-DA probe to analyze ROS levels by FACS analysis (**A**) after treatment with *R. roxburghii* fruit extract (CL-3) or vitamin C (Vc) for 48 h. The results of the quantitative analysis of (**A**) are shown in (**B**). PBS was used as a negative control (NC). (**C**) SOD activity was detected by a superoxide dismutase assay kit with NBT after *R. roxburghii* fruit extract treatment for 48 h. (**D**,**E**) Mesenchymal stem cells were pretreated with *R. roxburghii* fruit extract or vitamin C for 48 h and treated with 200 μΜ H_2_O_2_ for 4 h (**D**). Scal bar = 100 μm. The relative ROS level was detected by FACS analysis (**E**). Student’s *t*-test, * *p* < 0.05, ** *p* < 0.01, **** *p* < 0.0001.

**Figure 3 foods-13-01673-f003:**
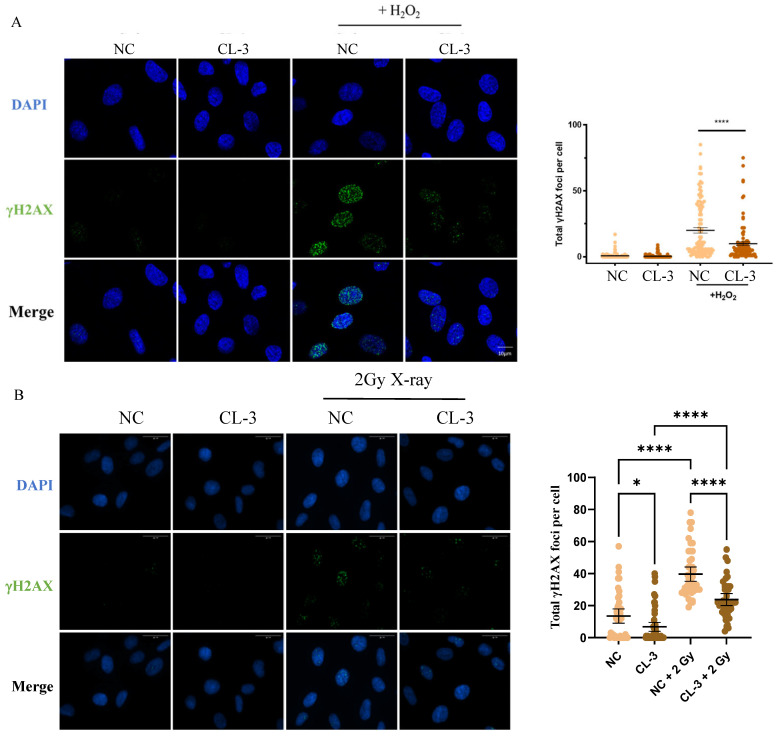
*R. roxburghii* fruit extract alleviates DNA damage in mesenchymal stem cells. (**A**) Mesenchymal stem cells were pretreated with *R. roxburghii* fruit extract for 24 h and then treated with 100 μM H_2_O_2_ for 2 h. The cells were fixed, and immunofluorescence was performed using a γH2AX antibody. γH2AX signals were statistically quantified in 100 cells. Scale bar = 10 μm. (**B**) Mesenchymal stem cells were pretreated with 100 μg/mL *R. roxburghii* fruit extract (CL-3) for 24 h and then treated with 2 Gy of X-rays. After 30 min, the cells were fixed, and immunofluorescence was detected using a γH2AX antibody. γH2AX signals were statistically quantified in 100 cells. The average fluorescence signal intensity was statistically analyzed. PBS was used as a negative control (Ctrl). Scale bar = 30 μm. One-way ANOVA was used for statistical analysis. * *p* < 0.05, **** *p* < 0.0001.

**Figure 4 foods-13-01673-f004:**
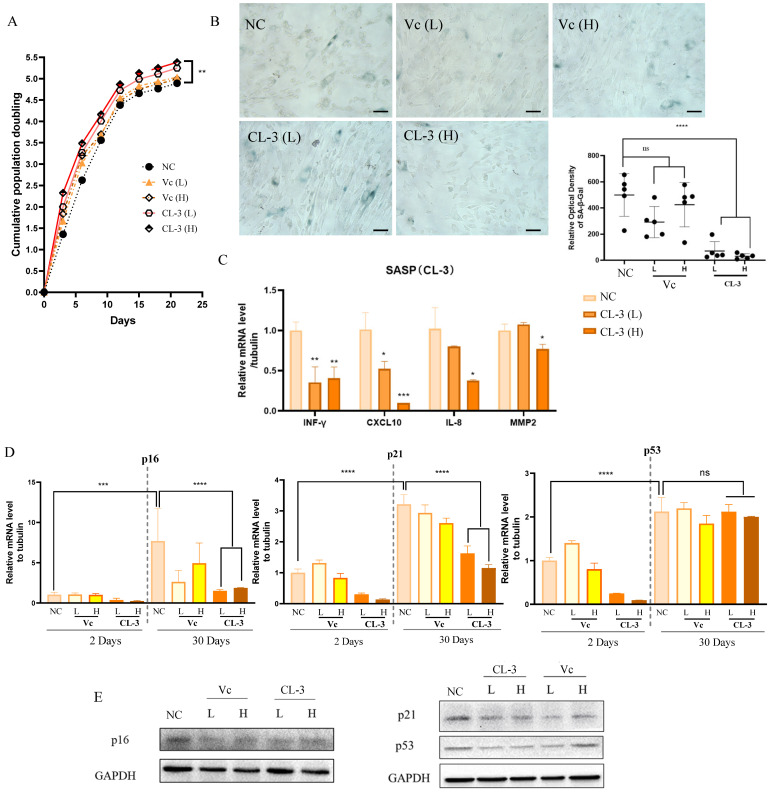
*R. roxburghii* fruit extract alleviates replicative senescence in mesenchymal stem cells. (**A**) Passage 6 mesenchymal stem cells were treated with *R. roxburghii* fruit extract or vitamin C every two days, cell numbers were counted, and cumulative population doubling was plotted. (**B**) SA-β-Gal staining of mesenchymal stem cells treated with *R. roxburghii* fruit extract or vitamin C for 30 days (left). Scale bar = 50 μm. The relative optical density of SA-β-Gal was quantified statistically. (**C**) mRNA levels of senescence-associated secreted protein (SASP) genes after 30 days of treatment. Student’s *t*-test, * *p* < 0.05, ** *p* < 0.01. (**D**) mRNA levels of p16, p21, and p53 after 2 days and 30 days of treatment. Student’s *t*-test, *** *p* < 0.001, **** *p* < 0.0001. (**E**) Protein levels of p16, p21, and p53 after 30 days of treatment.

**Figure 5 foods-13-01673-f005:**
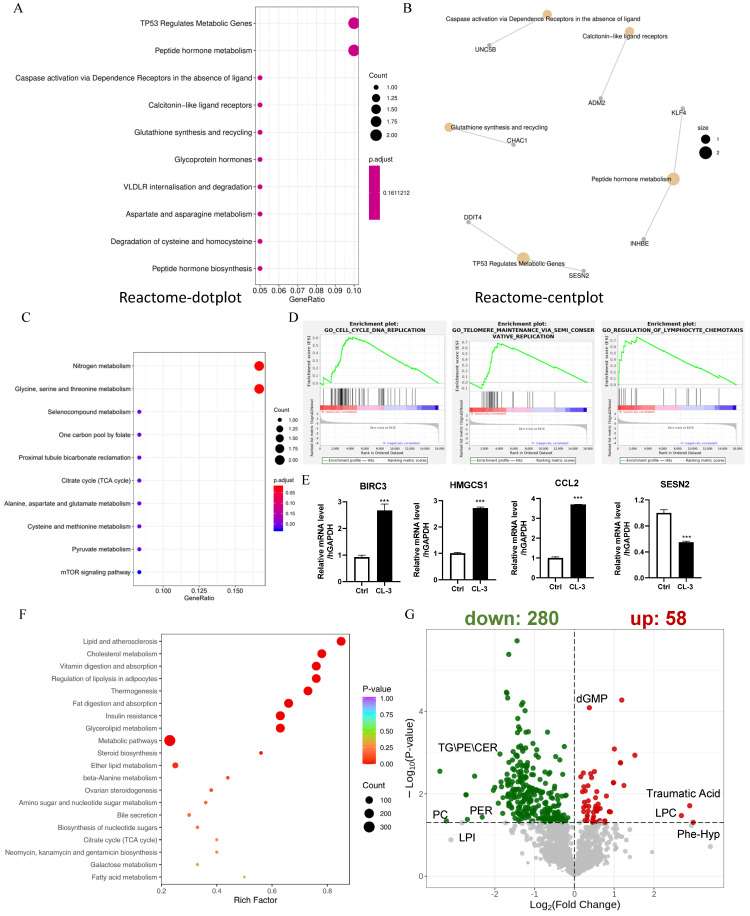
*R. roxburghii* fruit extract targets multiple pathways and metabolically modulates rejuvenation. (**A**,**B**) Reactome-dot plot and Reactome-cent plot of differentially expressed genes in *R. roxburghii* fruit extract-treated cells. (**C**) KEGG dot plot of differentially expressed genes in *R. `roxburghii* fruit extract-treated cells. (**D**) GSEA of differentially expressed genes in *R. roxburghii* fruit extract-treated cells showing DNA replication, telomere maintenance, and lymphocyte chemotaxis. (**E**) Real-time quantitative PCR of representative differentially expressed genes in *R. roxburghii* fruit extract-treated cells (CL-3). PBS served as a negative control (Ctrl). Student’s *t*-test, *** *p* < 0.001. (**F**) KEGG enrichment of differentially abundant metabolites in *R. roxburghii* fruit extract-treated cells. (**G**) Volcano plot of differentially abundant metabolites in *R. roxburghii* fruit extract-treated cells.

## Data Availability

The original contributions presented in the study are included in the article/Appendix A, further inquiries can be directed to the corresponding authors.

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
