# Peer review of "Rosa roxburghii Fruit Extracts Upregulate Telomerase Activity and Ameliorate Cell Replicative Senescence"

_foods, 2024, doi:10.3390/foods13111673_

Round 1

Reviewer 1 Report

Comments and Suggestions for Authors

In the submitted paper (foods-3021292), the authors have, for the first time, examined in detail, using a commendable set of methods, the potential of Rosa roxburghii plant extracts to upregulate telomerase activity and ameliorate cell replicative senescence in three commercial cell lines. As the authors rightly state, the obtained results can be a good starting point for further tests of the anti-ageing properties of the mixture of bioactive ingredients of this type of rose.

Taken comprehensively, this is a very interesting work on the current topic, well-conceived and executed, with well-presented and analyzed results. However, serious (primarily "technical") remarks must be addressed to the authors to polish the manuscript to a level worthy of publication. Let's go in order, one section at a time.

Abstract: All the usual parts are descriptively presented, but that's fine. What is the chemical nature (ie, which class of organic compounds does it belong to) of compound TA-65? This is not stated anywhere in the text!

Introduction: It's a bit unnecessarily long but very informative. Please (and everywhere else in the text) write Latin words and names in italics! For example, lines 53, 92, 96 etc.

Material and methods: Surprisingly, it is the weakest part of the text by far. Everything written is very good, with detailed descriptions of the procedures, except that sometimes the manufacturer's name or the device's model needs to be added (e.g. line 163). However, (all) analytical procedures are not written!!! It is an elementary mistake, enough to reject the paper, so I ask the authors to pay attention to it in future publications and to state everything that needs to be written in this article. For example, the most important thing is how the initial extracts were obtained for all further analyses, and this is not stated. What are the extracts in question, water, organic, combined? There are many such examples; of course, it is not enough to mention the commercial kit for determining SOD activity in the results and not write anything about it in this part (principle of the method, manufacturer, measurement wavelength, how the activity was calculated, whether the total SOD activity, since in cells CuZnSOD acts in the cytoplasm, and MnSOD acts in the mitochondria, etc.). Another example is the results talk about the protein levels of regulatory molecules, such as the tumour suppressor p16, p21 and p53 (Fig. 4E), and nowhere is it stated how this was done, of course, with the indication of whose (manufacturer) the antibodies were used in blot analyses. Simply put, this part must be reworked in detail, i.e. supplemented.

Results: Without substantive remarks, except that whenever it is mentioned in the text that there are statistically significant differences, the level of significance should also be indicated (in parentheses) (like line 292). A suggestion for further work (I will not take it here as aggravating) to the authors: it is necessary to do (at least partial) chemical characterization of the obtained extracts, not "blindly" guess what could cause the obtained (excellent) effects (lines 264-269).

Discussion: The second part of the study needs to be more robust. Not a single cited reference! Yes, it highlights the most important results and their significance, lists limitations (that's all been done), and compares the results and conclusions with the literature. Okay, the authors are the first to use plant extracts to examine the effect on telomerase activity as the main investigated parameter. Still, similar studies have been done with other plants/mixtures/compounds, such as TA-65, which must be mentioned and discussed here!

Finally, please correct minor errors, such as the formula (line 132), or replace the word drug (the authors did not work with any drugs) in a couple of places (e.g. lines 151 and 175). Also, check the English language; some sentences are grammatically incorrect or clumsy. For example, in the abstract, the fourth sentence should read: “Natural products can attenuate the aging process by increasing telomerase activity, such as TA-65.”

Comments on the Quality of English Language

The general quality of the English language should be better!

Reviewer 2 Report

Comments and Suggestions for Authors

See the attachment.

Reviewer 3 Report

Comments and Suggestions for Authors

Page 1 line 23 : what is the meaning of SOD.

page 1 line  30 : key words (Telomerase; Rosa roxburghii fruit; Cell senescence ,  ameliorate) instead of (Telomerase; Rosa roxburghii fruit; Cell senescence)

Page 5 line 216: What is the meaning of hMSC.

Page 13 lines 475-478: in conclusions, from my point of view, I see that the conclusion is short and abbreviated, which leads to a lack of clarity of the idea. Therefore, sufficient information needs to be added in order for the conclusion from the research to become clear.

Page 15 line 497 : the references need to update 

Comments on the Quality of English Language

 Minor editing of English language required

Round 2

Reviewer 1 Report

Comments and Suggestions for Authors

Congratulations to the authors on how efficiently they revised the text of their study. Now that everything is in its place, I will happily propose accepting this version of the paper.

Author Response

Thank you for your comments and all the suggestions.

Reviewer 2 Report

Comments and Suggestions for Authors

The authors satisfactorily responded to all suggestions. Discussion section could be additionally improved. 

Author Response

Thank you for your comments and we have revised the discussion again.